# Secure analysis in cognitive satellite-terrestrial networks with untrusted LEO relaying and friendly jamming

**Peng Zhang**⊕, **Jin Gao**⊕\* 

The School of Physics and Electronic Information, Weifang University, Weifang, Shandong, China

⊕ These authors contributed equally to this work.
\* gaojin_wfu@163.com

## Abstract

This study examines physical layer security in a cognitive satellite-terrestrial network where a ground user (GU) transmits confidential data to a Geostationary Earth Orbit (GEO) satellite via an untrusted Low Earth Orbit (LEO) relay adopting the Amplify-and-Forward (AF) relay strategy. To counter eavesdropping risks from the relaying LEO satellite, a friendly LEO jammer emits artificial noise. The secrecy rate is analyzed under cognitive radio constraints, considering interference thresholds, relay gain, and GU-LEO link shadowing. Numerical results show that increasing the interference threshold initially improves the secrecy rate but saturates when the maximum transmit power constraint at the GU is reached. When this maximum power constraint is removed, the interference threshold can be adjusted to optimize the secrecy rate. Additionally, we observe that lighter shadowing conditions generally enhance the secrecy rate, even though they improve the untrusted relay's channel. These findings offer valuable insights into optimizing security in cognitive satellite-terrestrial networks.

## Introduction

The integration of satellite and terrestrial networks has become an essential component in modern communication systems, particularly as demand for seamless, high-speed global coverage continues to grow [1–3]. Low Earth Orbit (LEO) satellites, due to their proximity to the Earth, offer low-latency and high-bandwidth capabilities, position them as ideal relay candidates in hybrid satellite-terrestrial networks [4]. LEO satellites have been also used as relay nodes to improve the connectivity between ground users (GUs) and Geostationary Earth Orbit (GEO) satellites, enabling efficient long-distance communications [5,6].

Recently, several studies (e.g., [7]) have investigated cognitive radio (CR) based hybrid satellite–terrestrial networks, in which the GU accesses the spectrum originally licensed to terrestrial primary users (PUs) to establish communication with satellites. This spectrum-sharing strategy is implemented to address the scarcity of available spectrum in satellite-terrestrial systems, allowing the GU to opportunistically access the same frequency bands as

**Data availability statement:** All relevant data are within the paper.

**Funding:** This Work is Supported by the National Natural Science Foundation of China

(61471269 and 61801266) and the Science and Technology Development Plan Project of Weifang City(2024GX027). The funders had no role in study design, data collection and analysis, decision to publish, or preparation of the manuscript.

**Competing interests:** The authors have declared that no competing interests exist.

terrestrial PUs. However, this introduces additional challenges, as the GU must limit its transmit power to avoid causing interference to PUs, such as terrestrial base stations and mobile users. Failure to control the interference can degrade the communication quality for the PUs. Consequently, power control becomes a critical component in ensuring delivery performance in both the primary and secondary networks, balancing the need for effective communication between the GU and the satellite while protecting the quality of service for the terrestrial PUs. This power control constraint also makes it challenging for the GU to establish a direct link to the GEO satellite, especially in scenarios with heavy obstructions. In such cases, LEO satellites can effectively serve as relays, forwarding messages from the GU to the GEO satellite [6]. Amplify-and-Forward (AF) relaying is particularly relevant in satellite systems due to its simplicity and efficiency in scenarios where the relay may not have the computational resources required for more complex strategies like Decode-and-Forward (DF) relaying [8].

However, with the increasing reliance on LEO satellites as relays, security concerns become even more pronounced when these relay satellites are considered untrusted. An untrusted LEO satellite may not only forward the GU's signal to the GEO satellite but may also attempt to eavesdrop on the transmitted information, posing a direct threat to the confidentiality of the communication. The challenge is further complicated by the fact that the untrusted LEO relay may have access to the codebook used by the GU, enabling it to decode the relayed messages. Moreover, any physical layer technique that aims to improve the connection between the GU and the GEO satellite, such as increasing the transmit power or optimizing signal transmission, will simultaneously enhance the connection between the GU and the LEO relay. This makes it challenging to secure the communication without inadvertently strengthening the eavesdropper's channel. Nevertheless, physical layer security techniques can still ensure secure communication by exploiting the inherent randomness of wireless channels [9,10], even if the untrusted relay has access to critical information, such as the codebook [11].

In recent years, there has been a growing body of research on physical layer security in terrestrial-satellite systems. For instance, in [12], the secrecy performance was investigated in a scenario involving randomly located vehicles distributed across different roads, where a satellite aims to securely serve one vehicle while preventing information leakage to another located elsewhere. In [13], the authors proposed a spatial cooperation strategy, wherein symbol mapping, beam steering, and antenna array partitioning were jointly optimized to confine the insecure region and suppress sidelobe radiation. The study in [14] considered the effect of terrestrial interference at the satellite and employed stochastic geometry to model the spatial distribution of LEO satellites, where multiple unintended LEO satellites attempt to eavesdrop on a legitimate communication link between a terrestrial IoT device and its intended LEO receiver. Moreover, artificial noise has also been investigated as a means to enhance communication secrecy. For example, [15] analyzed the security–reliability tradeoff in satellite–terrestrial relay networks assisted by a friendly jammer, where a terrestrial relay assists communication between a satellite and a ground user, while an external jammer broadcasts artificial noise to interfere with multiple nearby eavesdroppers.

More recently, physical layer security in cognitive satellite–terrestrial networks with artificial noise has also drawn attention. For instance, [16] studied the secrecy performance in the presence of artificial noise and non-colluding eavesdroppers. However, this work focused only on the secondary terrestrial network, while the security of the satellite–terrestrial network was ignored. Notably, scenarios involving cognitive satellite–terrestrial networks with untrusted LEO relays have received little attention and remain an open research direction. One of the key challenges lies in the analytical complexity introduced by terrestrial–satellite wireless channels. In most existing works (e.g., [12,15]), the derived expressions for performance metrics such as secrecy rate or secrecy outage probability are highly intricate, which

not only makes numerical evaluation computationally intensive but also hinders the extraction of useful design insights from the resulting formulas.

In this paper, we focus on analyzing the secrecy rate in a CR hybrid satellite-terrestrial network where an untrusted LEO satellite serves as an AF relay between a GU and a GEO satellite. In order to mitigate the eavesdropping threat posed by the untrusted LEO relay, a friendly LEO jammer transmits artificial noise to disrupt the relay's ability to intercept the communication. We derive closed-form expressions for the secrecy rate and investigate how different system parameters, such as channel fading, power control, and jamming, affect the physical layer security of the system. Our analysis is conducted within a CR framework, where the GU's transmit power is constrained by interference limits at terrestrial PUs. The key contributions of this paper are outlined as follows:

- We develop an analytical model for the secrecy rate in a CR satellite-terrestrial network with an untrusted LEO AF relay and a friendly LEO jammer, considering various system parameters such as channel fading and power constraints imposed by CR.
- We derive an exact closed-form expression for the secrecy rate, providing insights into the security performance of the system under untrusted AF relaying.
- We validate the analytical results through Monte Carlo simulations and explore how power limitations imposed by CR, and the impact of jamming and shadowing, affect the secrecy rate.

## System model

In this paper, we consider a cognitive satellite-terrestrial communication system, as shown in Fig 1, where a GU transmits confidential data to a GEO satellite through an untrusted LEO satellite, which operates as an AF relay. The DF protocol is not considered in this work, as it requires the relay to decode the legitimate user's message before forwarding [17]. In the untrusted relay scenario studied here, this would lead to full exposure of confidential information, making DF unsuitable. The system operates under a CR framework, meaning that the GU uses the same spectrum as the terrestrial primary networks. In such an environment, the transmit power of the GU is constrained to ensure that the interference to the terrestrial primary users (PUs), remains below a specific threshold. A direct connection between the GU and the GEO satellite is not feasible due to heavy obstructions, such as buildings or terrain, causing significant shadowing. Additionally, the power control imposed by the underlay CR framework further challenges the feasibility of a direct GU-to-GEO link, as the GU must limit its transmit power to avoid causing interference to terrestrial PUs.

The LEO relay is not fully trusted to forward the message securely, and it may potentially eavesdrop on the transmitted messages. To mitigate this risk, we introduce a cooperative jamming strategy where another LEO satellite serves as a friendly jammer. This jammer is located outside the field of view of the GU, so it cannot act as a possible relay for forwarding the messages from the GU to the GEO satellite. However, it can still transmit interference to the untrusted LEO relay, effectively degrading its ability to eavesdrop on the transmitted messages.

The message transmission from the GU to the GEO satellite occurs over two-time slots. In the first time slot, the GU transmits its signal to the untrusted LEO satellite, while a friendly LEO jammer simultaneously sends artificial noise to the untrusted LEO satellite to degrade its eavesdropping capability. In the second time slot, the untrusted LEO satellite forwards the signal received from the GU in the first time slot to the GEO satellite, applying the AF relaying strategy.

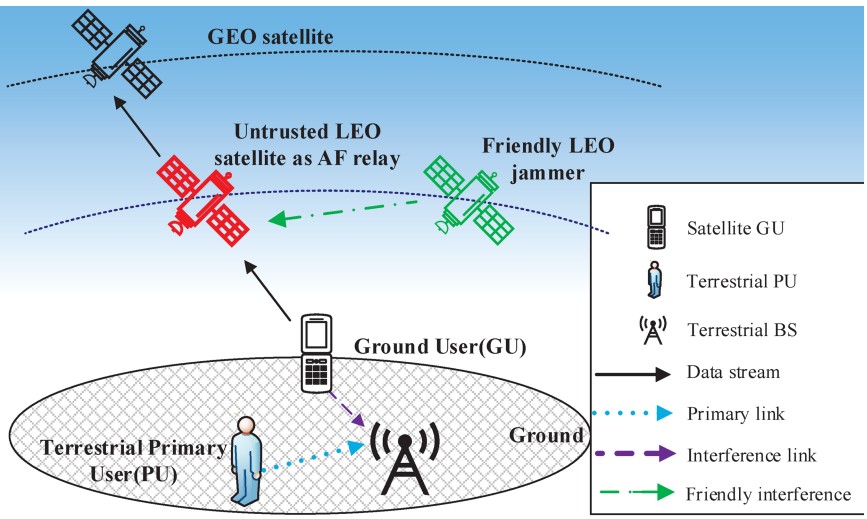

**Fig 1. System model.** A cognitive satellite-terrestrial communication system model.

We consider a widely used underlay CR framework, where the transmit power of the GU (denoted by $P_U$) is constrained to ensure that the interference at the PU remains below a pre-defined threshold, $I_{\text{th}}$. The channel gain between the GU and the PU is represented by $h_T$, and the interference constraint is expressed as $P_U|h_T|^2 \leq I_{\text{th}}$, Additionally, we account for a maximum power constraint, denoted by $P_M$, at the GU. Therefore, the transmit power at the GU in this underlay CR network is given by (cf. [18,19])

$$P_U = \min\left\{\frac{I_{\text{th}}}{|h_T|^2}, P_M\right\}. \tag{1}$$

In contrast to the GU's power control imposed by the CR framework, the friendly LEO jammer operates without such power limitations, as it is far away from the terrestrial primary networks.

## Communication links

In the following, we provide a detailed description of the channel models for the terrestrial-satellite link, the inter-satellite link, and the terrestrial link, respectively.

**GU-to-LEO link.** The GU transmits a signal to the LEO satellite over a wireless link subject to Rician-Shadowed fading. The channel gain for this link is denoted as $h_L$, and follows a Rician-Shadowed distribution characterized by three parameters: the parameter $m_0$ reflecting the average obstruction of the LOS component, the average power of the scattered component $2b$, and the average power of the LOS component, denoted by $\Omega$. The probability density function (PDF) of the Rician-Shadowed fading channel power gain $|h_L|^2$ is given by the following expression [20]

$$f_{|h_L|^2}(x) = \alpha \exp\left(-\beta x\right) {}_1F_1\left(m_0; 1; \delta x\right), \tag{2}$$

where $\alpha = \frac{(2bm_0)^{m_0}}{2b(2bm_0+\Omega)^{m_0}}$, $\beta = \frac{1}{2b}$, and $\delta = \frac{\Omega}{2b(2bm_0+\Omega)}$ are functions of the Rician-Shadowed fading parameters $m_0$, $b$, and $\Omega$, and ${}_1F_1(\cdot)$ is the confluent hypergeometric function of the first

kind. For simplification, we consider that $m_0$ is a positive integer. Then, the PDF and the cumulative distribution function (CDF) of $|h_L|^2$ are respectively of the form [12,21]

$$f_{|h_L|^2}(x) = \alpha \sum_{i=0}^{m_0-1} \zeta(i) x^i \exp(-(\beta - \delta)x), \tag{3}$$

$$F_{|h_L|^2}(x) = 1 - \alpha \sum_{i=0}^{m_0-1} \frac{\zeta(i)}{(\beta - \delta)^{i+1}} \Gamma(i + 1, (\beta - \delta)x), \tag{4}$$

where $\zeta(i) \triangleq \binom{m_0-1}{i} \frac{\delta^i}{i!}$, and $\Gamma(\cdot, \cdot)$ denotes the upper incomplete Gamma function.

**Satellite-to-satellite links.** The LEO satellite forwards the amplified signal to the GEO satellite. The LEO-to-GEO link is considered to be subject only to large-scale path loss, and the channel gain for this link is denoted as $L_G$, representing the path loss. Satellite-to-satellite links typically do not experience small-scale fading due to the absence of obstacles in space and the direct LoS nature of the communications, which eliminates the effects of multipath propagation. The signal received at the GEO satellite is given by

$$y_G = \sqrt{L_G} x_L + n_G, \tag{5}$$

where $x_L$ is the amplified signal sent from the LEO satellite, and $n_G$ is the Additive White Gaussian Noise (AWGN) at the GEO satellite with the variance of $\sigma_G^2$. Similar to the LEO-to-GEO link, we assume that the link between the LEO jammer and the untrusted LEO relay experiences only large-scale path loss (denoted by $L_J$), with no small-scale fading effects.

**GU-to-PU link.** For the GU-to-PU link, we assume a Nakagami-$m$ fading channel. The PDF and CDF of $|h_T|^2$, with a mean of $m\theta$, are expressed as follows

$$f_{|h_T|^2}(x) = \frac{1}{\Gamma(m)\theta^m} x^{m-1} \exp\left(-\frac{x}{\theta}\right), \tag{6}$$

$$F_{|h_T|^2}(x) = \frac{1}{\Gamma(m)} \Upsilon\left(m, \frac{x}{\theta}\right), \tag{7}$$

where $\Gamma(\cdot)$ represents the Gamma function, and $\Upsilon(\cdot, \cdot)$ denotes the lower incomplete Gamma function.

## Untrusted AF relaying

The untrusted LEO satellite employs the AF relaying strategy, in which it amplifies the received signal from the GU and forwards it to the GEO satellite. The amplification gain at the LEO satellite is denoted by $G_{AF}$, and the signal received at the GEO satellite after relaying is

$$y_G = \sqrt{L_G} G_{AF} \left(\sqrt{P_U} h_L x_U + n_L + \sqrt{L_J} z\right) + n_G, \tag{8}$$

where $x_U$ is the transmitted signal from the GU, $G_{AF}$ is the AF gain, and $n_L$ is the AWGN with the variance of $\sigma_L^2$ at the LEO satellite. In the above, $z \sim \mathcal{CN}(0, \sigma_Z^2)$ and $L_J$ represent the artificial noise and the transmit power of the friendly LEO jammer respectively. Notably, the artificial noise $z$ is known to the GEO satellite, whereas the untrusted LEO relay has no prior knowledge of it.

Since the LEO relay is untrusted, the GU does not perform channel training with the LEO satellite, meaning that the LEO satellite lacks knowledge of the instantaneous channel state information (CSI) for the GU-to-LEO link. This approach prevents the untrusted LEO relay from adapting its gain to degrade the SNR at the GEO satellite. In this scenario, a fixed amplification factor is considered, as the LEO satellite does not have the instantaneous CSI of the first hop. The fixed AF relay gain ensures that the LEO relay maintains a consistent level of amplification regardless of the instantaneous channel conditions between the GU and the LEO satellite. By fixing the relay gain, the system maintains control over potential security risks posed by the untrusted relay while enabling the LEO satellite to forward the signal to the GEO satellite.

Since the GU does not engage in channel training with the LEO satellite, and due to the absence of a direct transmission path between the GU and the GEO satellite caused by significant LoS obstructions, the GEO satellite lacks access to the instantaneous CSI for both the GU-LEO link and the GU-terrestrial PU link. As a result, the GEO satellite demodulates the received signal based solely on the statistical properties of the GU-LEO channel and the GU-PU channel. This reliance on statistical knowledge complicates the signal decoding process, as the GEO cannot utilize instantaneous CSI to optimize its decoding strategy. Instead, the GEO must rely on the channel's average behavior to decode the signal. Given this scenario and removing the artificial noise, the signal for decoding $x_U$ is of the form

$$y'_G = \sqrt{L_G}G_{\text{AF}}\left(\sqrt{P_U}h_L x_U + n_L\right) + n_G, \tag{9}$$

which can be rewritten as

$$\begin{aligned} y'_G = &\sqrt{L_G}G_{\text{AF}}\mathbb{E}\left\{\sqrt{P_U}|h_L|\right\}x_U \\ &+ \sqrt{L_G}G_{\text{AF}}\left(\sqrt{P_U}h_L - \mathbb{E}\left\{\sqrt{P_U}|h_L|\right\}\right)x_U \\ &+ \sqrt{L_G}G_{\text{AF}}n_L + n_G, \end{aligned} \tag{10}$$

where $\mathbb{E}\{\cdot\}$ denotes the expectation operator. The end-to-end SNR at the GEO satellite, denoted as $\gamma_G$, can be expressed as

$$\gamma_G = \frac{L_G G_{\text{AF}}^2 \mathbb{E}^2\left\{\sqrt{P_U}|h_L|\right\}}{\sigma_G^2 + L_G G_{\text{AF}}^2 \sigma_L^2 + L_G G_{\text{AF}}^2 \text{Var}\left\{\sqrt{P_U}h_L\right\}}, \tag{11}$$

where $\text{Var}\{\cdot\}$ denotes the variance of the inside random variable.

To decode the signal sent from the GU based on the statistical CSI knowledge, the received signal at the untrusted LEO satellite can be rewritten as

$$\begin{aligned} y_L = &\mathbb{E}\left\{\sqrt{P_U}|h_L|\right\}x_U \\ &+ \left(\sqrt{P_U}h_L - \mathbb{E}\left\{\sqrt{P_U}|h_L|\right\}\right)x_U \\ &+ \sqrt{L_J}z + n_L. \end{aligned} \tag{12}$$

The SNR for decoding $x_U$ at the LEO satellite is of the form

$$\gamma_L = \frac{\mathbb{E}^2\left\{\sqrt{P_U}|h_L|\right\}}{L_J \sigma_Z^2 + \sigma_L^2 + \text{Var}\left\{\sqrt{P_U}|h_L|\right\}}. \tag{13}$$

## Secrecy rate

We aim to analyze the secrecy rate of the communication link between the GU and the GEO satellite in the presence of both an untrusted LEO relay and a friendly LEO jammer. The secrecy rate is defined as the difference between the achievable rates at the GEO satellite and the LEO relay. This rate represents the secure transmission capacity of the information sent from the GU to the GEO satellite.

Mathematically, the secrecy rate $C_s$ is expressed as

$$C_s = \frac{1}{2} \Big[ \log_2(1 + \gamma_G) - \log_2(1 + \gamma_L) \Big]^+, \tag{14}$$

where $\gamma_G$ is the SNR at the GEO satellite, and $\gamma_L$ is the SNR at the untrusted LEO relay. The term $[x]^+ = \max\{x, 0\}$ ensures that the secrecy rate is non-negative. The friendly jammer reduces $\gamma_L$, thereby improving the secrecy rate by degrading the LEO relay's ability to intercept the message.

## Performance analysis

In this section, we will derive the closed-form expression for the secrecy rate. Observing (11), (13), and (14), we know that we should derive the closed-form expressions for $\mathbb{E}\{\sqrt{P_U}|h_L|\}$ and $\mathrm{Var}\{\sqrt{P_U}|h_L|\}$.

**Theorem 1.** *In the presence of an untrusted LEO relay and an LEO jammer, the secrecy rate of the considered hybrid cognitive terrestrial-satellite networks takes the form in* (14) *with the expressions for $\mathbb{E}\{\sqrt{P_U}|h_L|\}$ and $\mathrm{Var}\{\sqrt{P_U}|h_L|\}$ respectively given by*

$$\mathbb{E}\{\sqrt{P_U}|h_L|\} = \mathbb{E}\{\sqrt{P_U}\}\mathbb{E}\{|h_L|\} \tag{15}$$

$$\mathrm{Var}\{\sqrt{P_U}|h_L|\} = \mathbb{E}\{P_U\}\mathbb{E}\{|h_L|^2\} - \mathbb{E}^2\{\sqrt{P_U}\}\mathbb{E}^2\{|h_L|\} \tag{16}$$

*where the mean of $\sqrt{P_U}$ is given by*

$$\mathbb{E}\{\sqrt{P_U}\} = \frac{\sqrt{P_M}}{\Gamma(m)} \Upsilon\left(m, \sqrt{\frac{I_{\mathrm{th}}}{P_M \theta^2}}\right) + \frac{\sqrt{I_{\mathrm{th}}/\theta}}{\Gamma(m)} \Gamma\left(m - 0.5, \frac{I_{\mathrm{th}}}{P_M \theta}\right). \tag{17}$$

*The closed-form expressions for $\mathbb{E}\{\sqrt{P_U}\}$, $\mathbb{E}\{|h_L|\}$ and $\mathbb{E}\{|h_L|^2\}$ are respectively*

$$\mathbb{E}\{P_U\} = \frac{P_M}{\Gamma(m)} \Upsilon\left(m, \frac{I_{\mathrm{th}}}{P_M \theta}\right) + \frac{I_{th}/\theta}{\Gamma(m)} \Gamma\left(m - 1, \frac{I_{\mathrm{th}}}{P_M \theta}\right), \tag{18}$$

$$\mathbb{E}\{|h_L|\} = \alpha \sum_{i=0}^{m_0 - 1} \zeta(i) \frac{\Gamma(i + 1.5)}{(\beta - \delta)^{i + 1.5}}, \tag{19}$$

$$\mathbb{E}\{|h_L|^2\} = 2b + \Omega. \tag{20}$$

*Proof of Theorem 1*: For $\mathbb{E}\{\sqrt{P_U}\}$, we have

$$\mathbb{E}\{\sqrt{P_U}\} = \int_0^\infty \sqrt{P_U} f_{|h_T|^2}(x) \mathrm{d}x$$

$$= \int_0^\infty \min\left\{\sqrt{\frac{I_{\mathrm{th}}}{x}}, \sqrt{P_M}\right\} f_{|h_T|^2}(x) \mathrm{d}x$$

$$= \int_0^{I_{\text{th}}/P_M} \sqrt{P_M} f_{|h_T|^2}(x)\mathrm{d}x + \int_{I_{\text{th}}/P_M}^{\infty} \sqrt{\frac{I_{th}}{x}} f_{|h_T|^2}(x)\mathrm{d}x$$

$$= \sqrt{P_M} F_{|h_T|^2}\left(\frac{I_{\text{th}}}{P_M}\right) + \int_{I_{\text{th}}/P_M}^{\infty} \sqrt{\frac{I_{th}}{x}} f_{|h_T|^2}(x)\mathrm{d}x. \tag{21}$$

For the second term in the above, we have

$$\int_{I_{\text{th}}/P_M}^{\infty} \sqrt{\frac{I_{th}}{x}} f_{|h_T|^2}(x)\mathrm{d}x = \frac{\sqrt{I_{\text{th}}}}{\Gamma(m)\theta^m} \int_{I_{\text{th}}/P_M}^{\infty} x^{m-1.5} \exp\left(-\frac{x}{\theta}\right)\mathrm{d}x$$

$$\overset{(a)}{=} \frac{\sqrt{I_{\text{th}}/\theta}}{\Gamma(m)}\Gamma\left(m-0.5, \frac{I_{\text{th}}}{P_M\theta}\right) \tag{22}$$

where (a) follows from [22, Eq. (3.381.3)].

For the mean of $P_U$, we have

$$\mathbb{E}\{P_U\} = \int_0^{\infty} \min\left\{\frac{I_{\text{th}}}{x}, P_M\right\} f_{|h_T|^2}(x)\mathrm{d}x$$

$$= \int_0^{I_{\text{th}}/P_M} P_M f_{|h_T|^2}(x)\mathrm{d}x + \int_{I_{\text{th}}/P_M}^{\infty} \frac{I_{th}}{x} f_{|h_T|^2}(x)\mathrm{d}x$$

$$= P_M F_{|h_T|^2}\left(\frac{I_{\text{th}}}{P_M}\right) + \int_{I_{\text{th}}/P_M}^{\infty} \frac{I_{th}}{x} f_{|h_T|^2}(x)\mathrm{d}x \tag{23}$$

where

$$\int_{I_{\text{th}}/P_M}^{\infty} \frac{I_{th}}{x} f_{|h_T|^2}(x)\mathrm{d}x = \frac{I_{th}}{\Gamma(m)\theta^m} \int_{I_{\text{th}}/P_M}^{\infty} x^{m-2} \exp(-x/\theta)\mathrm{d}x$$

$$\overset{(a)}{=} \frac{I_{th}/\theta}{\Gamma(m)}\Gamma\left(m-1, \frac{I_{\text{th}}}{P_M\theta}\right), \tag{24}$$

where (a) follows from [22, Eq. (3.381.3)]. Combining (23) and (24) yields (18).

For $\mathbb{E}\{|h_L|\}$, by using the PDF of $|h_L|^2$, we have

$$\mathbb{E}\{|h_L|\} = \alpha \sum_{i=0}^{m_0-1} \zeta(i) \int_0^{\infty} x^{i+0.5} \exp(-(\beta-\delta)x)\mathrm{d}x$$

$$\overset{(a)}{=} \alpha \sum_{i=0}^{m_0-1} \zeta(i) \frac{\Gamma(i+1.5)}{(\beta-\delta)^{i+1.5}} \tag{25}$$

where (a) follows from [22, Eq. (3.381.4)]. Referring to [23, Prop. 5.1], we know that

$$\mathbb{E}\{|h_L|^2\} = 2b + \Omega. \tag{26}$$

We conclude the proof. □

**Remark 1.** Theorem 1 introduces a novel analytical framework for characterizing the secrecy rate in the considered system. Notably, the proposed framework can potentially be extended to address other physical-layer security concerns, such as covert communications [24], where the untrusted relay is unaware of the legitimate user's transmission and simply amplifies and forwards the received signal.

## Numerical results and discussion

In this section, we present the numerical results for the secrecy rate in the considered cognitive satellite-terrestrial network with an untrusted LEO relay and a friendly LEO jammer. The results are validated through Monte Carlo simulations, and we analyze the impact of various system parameters on the secrecy rate. For simplicity, we set $L_J = L_G = \sigma_G^2 = \sigma_L^2 = \theta = 1$. In Figs 2–4 , the GU-to-LEO relay link is subject to infrequent light shadowing, with parameters $m_0 = 20$, $b = 0.158$, and $\Omega = 1.29$. In Fig 5, we examine three different shadowing conditions for the GU-to-LEO relay link: frequent heavy shadowing ($m_0 = 1$, $b = 0.063$, and $\Omega = 8.97 \times 10^{-4}$), average shadowing ($m_0 = 10$, $b = 0.126$, and $\Omega = 0.835$), and infrequent light shadowing ($m_0 = 20$, $b = 0.158$, and $\Omega = 1.29$). The parameter settings for the three shadowing scenarios are summarized in Table 1.

Fig 2 illustrates the impact of the interference threshold $I_{th}$ on the secrecy rate under different artificial noise power levels. As $I_{th}$ increases, the secrecy rate improves, as a higher threshold allows the GU to transmit with greater power, enhancing the SNR at the GEO satellite. However, beyond a certain point, the secrecy rate stabilizes and becomes nearly unchanged. This is due to the maximum transmit power constraint at the GU, which limits the GU's ability to further increase its power, regardless of the increase in $I_{th}$. Therefore, once the GU reaches its maximum transmit power, further increases in $I_{th}$ provide no additional benefit. This indicates that the system is bound by the maximum power limit of the GU, making the adjustment of $I_{th}$ effective only up to a certain level. Beyond that, the secrecy rate plateaus, and the system performance depends more on other factors, such as the artificial noise power $\sigma_Z^2$.

In Fig 3, we remove the maximum transmit power constraint, allowing the GU to transmit with power always close to $I_{th}/|h_T|^2$. Under this setting, the secrecy rate initially increases with $I_{th}$, reaching a peak, and then decreases. The initial increase occurs because a higher $I_{th}$

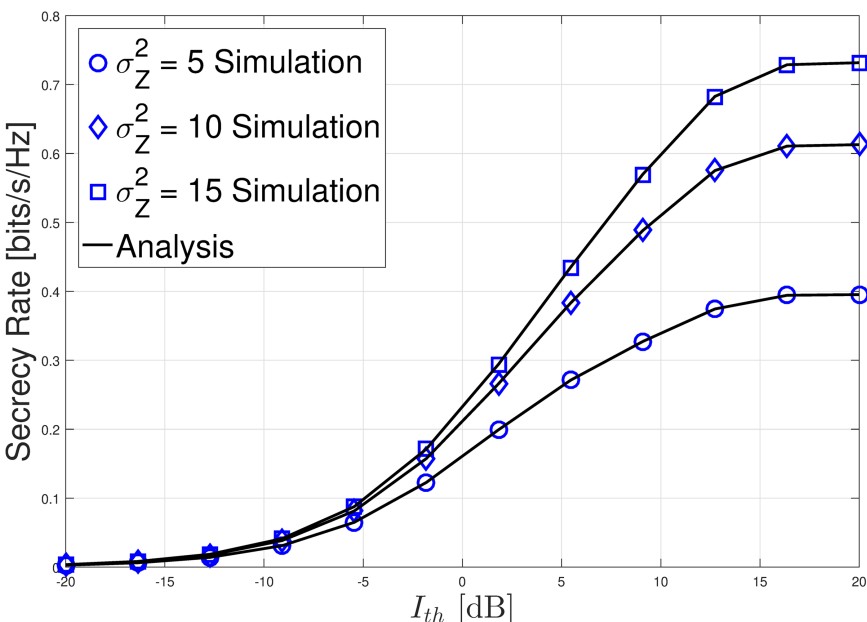

**Fig 2. Secrecy rate versus $I_{th}$ with $P_M = 10$ dB, $G_{AF} = 1$, $m = 2$, $m_0 = 20$, $b = 0.158$, and $\Omega = 1.29$.** The impact of the interference threshold $I_{th}$ on the secrecy rate under different artificial noise power levels.

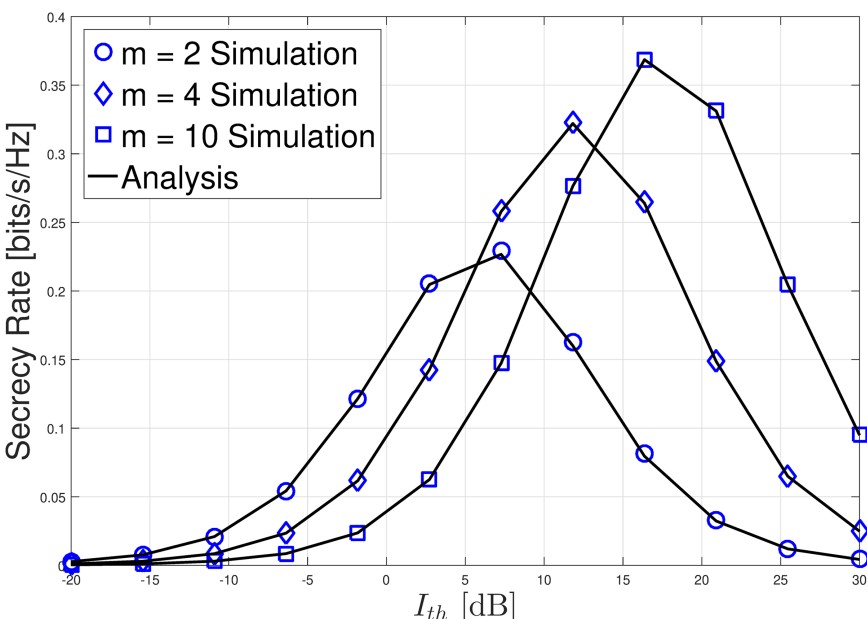

**Fig 3. Secrecy rate versus $I_{\text{th}}$ with $P_M \to \infty$, $G_{\text{AF}} = 1$, $\sigma_Z^2 = 5$, $m_0 = 20$, $b = 0.158$, and $\Omega = 1.29$.**

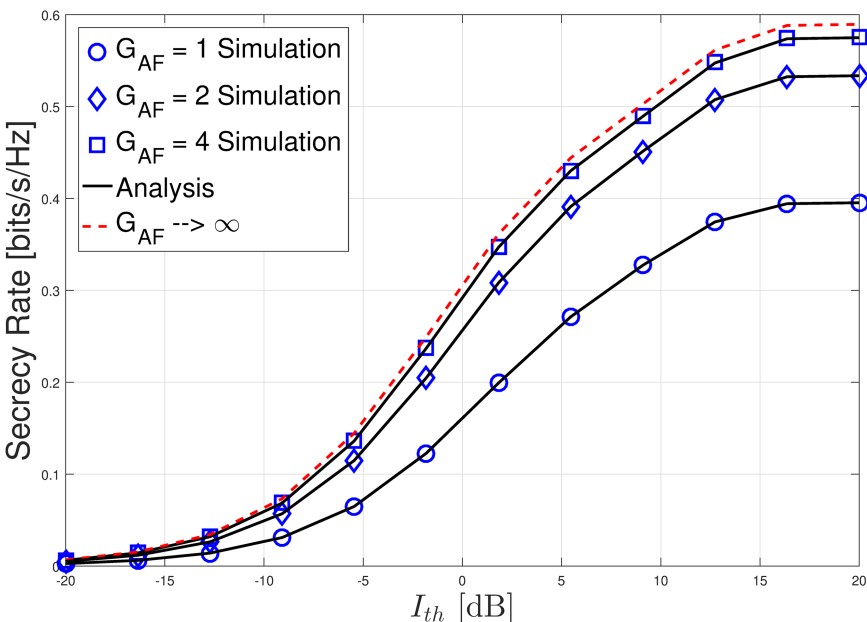

**Fig 4. Secrecy rate versus $I_{\text{th}}$ with $P_M = 10$ dB, $m = 2$, $m_0 = 20$, $b = 0.158$, and $\Omega = 1.29$.**

enhances the link quality between the GU and the GEO satellite, improving the achievable rate at the GEO satellite. However, as $I_{\text{th}}$ continues to increase, the untrusted LEO relay's ability to intercept the message improves as well, because the influence of the artificial noise $\sigma_Z^2$ becomes less significant. When $I_{\text{th}}$ is sufficiently large, the artificial noise becomes almost

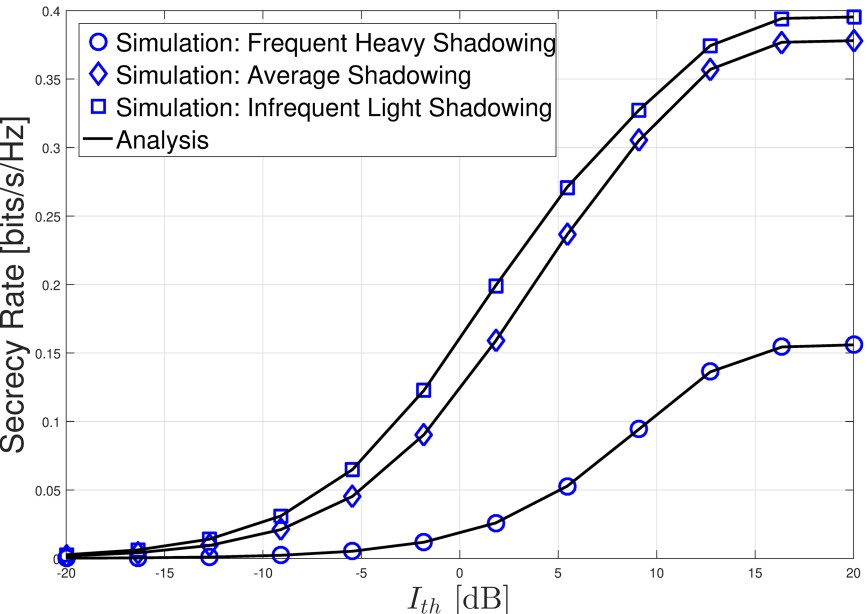

**Fig 5. Secrecy rate versus $I_{th}$ with $P_M$ = 10 dB, $G_{AF}$ = 1, $m$ = 2 and $\sigma_Z^2$ = 5.**

**Table 1. Parameters in three typical shadowing scenarios [20, Table III].**

| Scenarios | $m_0$ | $b$ | $\Omega$ |
|---|---|---|---|
| Frequent Heavy Shadowing | 1 | 0.063 | $8.97 \times 10^{-4}$ |
| Average Shadowing | 10 | 0.126 | 0.835 |
| Infrequent Light Shadowing | 20 | 0.158 | 1.29 |

ineffective at hindering the untrusted relay, leading to a decrease in the secrecy rate. This suggests that there is an optimal value of $I_{th}$ that balances the improvement in the legitimate link and the negative effect on the eavesdropping link. The value of $I_{th}$ can be adjusted by varying the distance between the GU and the PU, which is a crucial factor in determining the overall secrecy performance.

Fig 4 examines the effect of the AF gain $G_{AF}$ on the secrecy rate. As $G_{AF}$ increases, the secrecy rate also improves because a higher AF gain amplifies the received signal at the GEO satellite, leading to a better end-to-end SNR. However, as $G_{AF}$ becomes sufficiently large, further increases no longer result in noticeable improvements in the secrecy rate. This is due to the fact that a higher AF gain also amplifies the noise in the first hop (GU-to-LEO link). As a result, while the legitimate signal is amplified, the accompanying noise is also magnified, which limits the benefit of increasing $G_{AF}$ indefinitely. The results indicate that although a moderate AF gain can significantly improve the secrecy rate, there is a saturation point beyond which the noise amplification "equals" the signal amplification.

Fig 5 shows the secrecy rate under three different shadowing conditions for the GU-to-LEO link: frequent heavy shadowing, average shadowing, and infrequent light shadowing. As expected, lighter shadowing leads to higher SNRs at both the untrusted LEO relay and the GEO satellite. While lighter shadowing improves the decoding capability of the untrusted relay, it also benefits the legitimate link to the GEO satellite by improving the end-to-end SNR. Interestingly, the results imply that lighter shadowing is generally favorable for secure

transmission, as the overall improvement in the end-to-end link dominates the potential advantage gained by the untrusted relay. Therefore, despite the improvement in the eavesdropper's channel quality, the GEO satellite's ability to decode the confidential message is enhanced more substantially under lighter shadowing conditions, leading to better secrecy performance.

In conclusion, the numerical results reveal that key parameters such as the interference threshold $I_{th}$, AF gain $G_{AF}$, and the shadowing conditions play significant roles in determining the secrecy rate. Adjusting these parameters can help balance the improvement in the legitimate link and the impact of eavesdropping, providing valuable insights into how to optimize the system's secrecy performance in the presence of an untrusted LEO relay and a friendly LEO jammer. Moreover, the excellent agreement between the simulated results and the theoretical analysis in Figs 2–5 fully validates the accuracy and reliability of the developed analytical model, confirming its robustness in predicting system behavior under various conditions.

## Conclusions

In this paper, we analyzed the physical layer security of a cognitive hybrid satellite-terrestrial communication system where a ground user communicates with a GEO satellite through an untrusted LEO relay, with the presence of a friendly LEO jammer. We derived the closed-form expression for the secrecy rate and investigated the impact of key system parameters, such as the interference threshold, AF gain, and shadowing effects, on the secrecy performance. Our results show that increasing the interference threshold improves the secrecy rate up to a certain limit, after which the impact of the untrusted relay diminishes the overall security. The AF gain was also found to improve secrecy but with diminishing returns as the noise amplification limits its benefits. Lastly, our analysis revealed that lighter shadowing conditions, despite improving the untrusted relay's channel, generally favor secure communication by significantly enhancing the end-to-end SNR at the GEO satellite.

The findings highlight the importance of carefully balancing system parameters to optimize security in cognitive satellite-terrestrial networks. Future work can explore further optimizations, such as power allocation strategies and advanced jamming techniques, to enhance the physical layer security in these systems.

## Author contributions

**Conceptualization:** Peng Zhang.

**Formal analysis:** Peng Zhang, Jin Gao.

**Investigation:** Peng Zhang, Jin Gao.

**Methodology:** Peng Zhang, Jin Gao.

**Resources:** Jin Gao.

**Software:** Jin Gao.

**Validation:** Jin Gao.

**Visualization:** Jin Gao.

**Writing – original draft:** Peng Zhang.

**Writing – review & editing:** Jin Gao.

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
