## [Decision Letter · Decision Letter 0]

9 May 2025

PONE-D-25-16284Secure analysis in cognitive satellite-terrestrial networks with untrusted LEO relaying and friendly jammingPLOS ONE

Dear Dr. Gao,

Thank you for submitting your manuscript to PLOS ONE. After careful consideration, we feel that it has merit but does not fully meet PLOS ONE’s publication criteria as it currently stands. Therefore, we invite you to submit a revised version of the manuscript that addresses the points raised during the review process.

We look forward to receiving your revised manuscript.

Kind regards,

Neng Ye

Academic Editor

PLOS ONE

6. We notice that your supplementary figures are uploaded with the file type 'Figure'. Please amend the file type to 'Supporting Information'. Please ensure that each Supporting Information file has a legend listed in the manuscript after the references list.

Additional Editor Comments (if provided):

Reviewers' comments:

Reviewer's Responses to Questions

**Comments to the Author**

1. Is the manuscript technically sound, and do the data support the conclusions?

Reviewer #1: Yes

Reviewer #2: Yes

Reviewer #3: Partly

2. Has the statistical analysis been performed appropriately and rigorously? 

Reviewer #1: Yes

Reviewer #2: N/A

Reviewer #3: Yes

3. Have the authors made all data underlying the findings in their manuscript fully available?

Reviewer #1: Yes

Reviewer #2: Yes

Reviewer #3: Yes

4. Is the manuscript presented in an intelligible fashion and written in standard English?

Reviewer #1: Yes

Reviewer #2: Yes

Reviewer #3: Yes

5. Review Comments to the Author

Reviewer #1: It is highly recommended to enrich the references and consider more recent, 2024-2025, publications on the field. You should study more update papers and compare them with your achievement for the sake of more acceptable presentation.

Reviewer #2: Review of the article "Secure analysis in cognitive satellite-terrestrial networks with untrusted LEO relaying and friendly jamming" submitted to the PLOS ONE

1. The article aligns well with the journal's scope and is presented in an organized manner, making it easy to read and follow.

2. The authors have investigated physical layer security in a cognitive satellite-terrestrial network where a ground user transmits confidential data to a Geostationary Earth Orbit satellite via an untrusted Low Earth Orbit relay, adopting the Amplify-and-Forward relay strategy.

3. A friendly LEO jammer generates artificial noise to mitigate eavesdropping risks from the relaying LEO satellite. The secrecy rate is assessed under cognitive radio constraints, considering interference thresholds, relay gain, and GU-LEO link shadowing.

4. Numerical results indicate that raising the interference threshold enhances the secrecy rate but levels off once the maximum transmit power constraint at the GU is reached.

5. The authors have concluded that when the maximum power constraint is removed, the interference threshold can be adjusted to optimize the secrecy rate. Additionally, lighter shadowing conditions enhance the secrecy rate, even though they improve the channel conditions for the untrusted relay.

6. The methodology is clearly presented, with extensive explanations and discussions of both the method and the results.

However, I suggest the authors address the following issues:

1. The authors need to compare the proposed analysis's performance with that of existing literature to demonstrate its robustness.

2. I suggest the authors slightly revise the language of the article. The attached file includes many suggestions.

Reviewer #3: The manuscript entitled “Secure analysis in cognitive satellite-terrestrial networks with untrusted LEO relaying and friendly jamming” investigates the secrecy rate in cognitive-radio hybrid satellite-terrestrial networks and is well organized with dedicated sections on technological overviews. Here are some issues that the reviewer conserns.

1. The Introduction spends too much time detailing the general pros and cons of LEO satellites for communication without critically analyzing recent works on secure communications. It fails to highlight the key challenges and advances in the field.

2. In “Numerical Results and Discussion,” the manuscript thoroughly examines how different artificial-noise power levels and shadowing conditions affect the secrecy rate, but relies solely on Monte Carlo simulations. It is recommended to include comparisons with data from actual communication scenarios.

3. The Conclusions mention only further parameter optimization under the amplify-and-forward scheme. The authors should also consider whether their closed-form expressions can be applied to alternative security strategies, such as decode-and-forward relaying.

4. Regarding the secure-capacity analysis in the manuscript�the authors may expand their research to some recent studies, such as “Achieving Positive Rate of Covert Communications Covered by Randomly Activated Overt Users” derives a tighter upper bound on the total variation distance for randomly activated overt users in both single-frame and multi-frame transmission scenarios�as well as spatially averaged secure capacity in "Cooperative Secure Transmission for Satellite Downlink: A Pas de Deux between Beam Pointings", etc. The authors may refer to these papers to improve their research.

6. PLOS authors have the option to publish the peer review history of their article (what does this mean?). If published, this will include your full peer review and any attached files.

Reviewer #1: No

Reviewer #2: **Yes: **Jawad K. Ali

Reviewer #3: No

---

## [Author Response · Author response to Decision Letter 1]

24 Jun 2025

First of all, we would like to sincerely thank you for your time and effort in handling the review process of our manuscript.

In the revised version, we have highlighted all major modifications in blue for your convenience. Based on the valuable suggestions and comments, the primary changes are summarized as follows:

1�We have substantially rewritten the Introduction section to critically analyze recent works on secure terrestrial-satellite communications. We have also clearly pointed out how our work differs from existing studies and highlighted the resulting technical challenges.

2�We have provided a detailed explanation of why the decode-and-forward (DF) relaying strategy, although potentially offering better relay performance, is not applicable under the considered system model. Additionally, we have discussed the feasibility of extending our analytical model to future studies on covert communications.

For detailed clarifications on each point, please refer to the document 'Response to Reviewers.pdf'. We once again express our sincere gratitude for the insightful feedback from both the Editor and the Reviewers, which helped us improve the quality of the manuscript.

Yours Sincerely,

Peng Zhang and Jin Gao

---

## [Decision Letter · Decision Letter 1]

6 Jul 2025

Secure analysis in cognitive satellite-terrestrial networks with untrusted LEO relaying and friendly jamming

PONE-D-25-16284R1

Dear Dr. Gao,

We’re pleased to inform you that your manuscript has been judged scientifically suitable for publication and will be formally accepted for publication once it meets all outstanding technical requirements.

Kind regards,

Neng Ye

Academic Editor

PLOS ONE

Additional Editor Comments (optional):

Reviewers' comments:

Reviewer's Responses to Questions

**Comments to the Author**

1. If the authors have adequately addressed your comments raised in a previous round of review and you feel that this manuscript is now acceptable for publication, you may indicate that here to bypass the “Comments to the Author” section, enter your conflict of interest statement in the “Confidential to Editor” section, and submit your "Accept" recommendation.

Reviewer #2: All comments have been addressed

Reviewer #3: All comments have been addressed

2. Is the manuscript technically sound, and do the data support the conclusions?

Reviewer #2: Yes

Reviewer #3: Yes

3. Has the statistical analysis been performed appropriately and rigorously? 

Reviewer #2: N/A

Reviewer #3: Yes

4. Have the authors made all data underlying the findings in their manuscript fully available?

Reviewer #2: Yes

Reviewer #3: Yes

5. Is the manuscript presented in an intelligible fashion and written in standard English?

Reviewer #2: Yes

Reviewer #3: Yes

6. Review Comments to the Author

Reviewer #2: Thanks to the authors for the detailed response and additions. I read through the comments and skimmed the revised version. The updates improved the paper a lot. I would be happy to recommend this paper for publication.

Reviewer #3: The authors have thoroughly addressed all the previous comments. I have no further concerns and recommend acceptance of this version.

7. PLOS authors have the option to publish the peer review history of their article (what does this mean?). If published, this will include your full peer review and any attached files.

Reviewer #2: **Yes: **Jawad K. Ali

Reviewer #3: No

---

## [Editor Report · Acceptance letter]

PONE-D-25-16284R1

PLOS ONE

Dear Dr. Gao,

I'm pleased to inform you that your manuscript has been deemed suitable for publication in PLOS ONE. Congratulations! Your manuscript is now being handed over to our production team.

Kind regards,

on behalf of

Dr. Neng Ye

Academic Editor

PLOS ONE